# Sharpless Asymmetric Dihydroxylation: An Impressive Gadget for the Synthesis of Natural Products: A Review

**DOI:** 10.3390/molecules28062722

**Published:** 2023-03-17

**Authors:** Aqsa Mushtaq, Ameer Fawad Zahoor, Muhammad Bilal, Syed Makhdoom Hussain, Muhammad Irfan, Rabia Akhtar, Ali Irfan, Katarzyna Kotwica-Mojzych, Mariusz Mojzych

**Affiliations:** 1Medicinal Chemistry Research Lab, Department of Chemistry, Government College University Faisalabad, Faisalabad 38000, Pakistan; 2College of Computer Science and Technology, Zhejiang University, Hangzhou 310027, China; 3Department of Zoology, Government College University Faisalabad, Faisalabad 38000, Pakistan; 4Department of Pharmaceutics, Government College University Faisalabad, Faisalabad 38000, Pakistan; 5Department of Chemistry, Superior University, Faisalabad 38000, Pakistan; 6Laboratory of Experimental Cytology, Medical University of Lublin, Radziwiłłowska 11, 20-080 Lublin, Poland; 7Department of Chemistry, Siedlce University of Natural Sciences and Humanities, 3-Go Maja 54, 08-110 Siedlce, Poland

**Keywords:** Sharpless asymmetric dihydroxylation, natural products, enantioselective, alkaloids, lactones, flavones, macrolides, polyketides

## Abstract

Sharpless asymmetric dihydroxylation is an important reaction in the enantioselective synthesis of chiral vicinal diols that involves the treatment of alkene with osmium tetroxide along with optically active quinine ligand. Sharpless introduced this methodology after considering the importance of enantioselectivity in the total synthesis of medicinally important compounds. Vicinal diols, produced as a result of this reaction, act as intermediates in the synthesis of different naturally occurring compounds. Hence, Sharpless asymmetric dihydroxylation plays an important role in synthetic organic chemistry due to its undeniable contribution to the synthesis of biologically active organic compounds. This review emphasizes the significance of Sharpless asymmetric dihydroxylation in the total synthesis of various natural products, published since 2020.

## 1. Introduction

Asymmetric synthesis plays an important role in the stereoisomeric and enantiomeric synthesis of drugs [1]. In this regard, Sharpless in the 1980s introduced Sharpless epoxidation, which is a facile methodology for the asymmetric synthesis of 2,3-epoxy alcohols by using primary and secondary allylic alcohols [2]. Considering the importance of enantioselective synthesis of Sharpless epoxidation, Sharpless asymmetric dihydroxylation was then employed by Sharpless to synthesize vicinal diols [3,4]. In this methodology, chiral vicinal diol moiety is obtained by the reaction of an alkene with osmium tetroxide in the presence of an optically active quinine ligand [5,6].

The presence of a chiral ligand is the primary requirement for osmium catalyzed bishydroxylation [7]. Since the inception of this protocol, various co-oxidant ligands have been employed by different researchers. However, it was inferred that low yields of diol were obtained by employing stoichiometric oxidants, i.e., H_2_O_2_ and NaClO_3_ or KClO3 [8,9]. Improvements in the yield of target molecules were observed by employing Upjohn dihydroxylation in the presence of *N*-methylmorpholine *N*-oxide and by using alkaline tert-butyl hydroperoxide (t-BHP) [10]. Moreover, potassium hexacyanoferrate (III) has been found to be the most effective in recent times. Similarly, Sharpless also utilized this optically active terminal oxidant for the bishydroxylation of alkenes [11]. As time passed, Sharpless and his coworkers devoted their attention to the use of cinchona alkaloids as chiral ligands, which gave tremendous yields with high enantioselectivity. Later on, pyrimidine and phthalazine incorporated dimeric alkaloids were employed as asymmetric ligands for the efficient synthesis of vicinal diols. K_2_OsO_2_(OH)_4_, K_2_CO_3_, (DHQD)_2_PHAL, (DHQ)_2_PHAL, and K_3_Fe(CN)_6_ were found to be mandatory chemical substances for carrying out Sharpless asymmetric dihydroxylation. The mixture of these four reagents is referred to as “AD-mix”. The (DHQ)_2_PHAL-containing mixture is termed “AD-mix-α”, while “AD-mix-β” includes (DHQD)_2_PHAL as ligand [11,12]. The general scheme for osmium-catalyzed asymmetric dihydroxylation is given below [12] (Figure 1).

Sharpless asymmetric dihydroxylation has found tremendous applications in the synthesis of a variety of naturally occurring biologically active compounds [13,14]. Total synthesis of various natural products such as alkaloids, lactones, amino acids, flavones, polyketides, macrolides, glycosides, terpenes, etc., involves Sharpless asymmetric dihydroxylation as an integral step [15,16,17,18]. This protocol results in the high enantioselectivity of vicinal diols, which are then reacted further in the presence of required reagents to synthesize the desired medicinally important target molecules [19,20]. The following figure illustrates the structure of a few biologically important natural products, whose total synthesis is achieved by employing Sharpless asymmetric dihydroxylation as an essential step (Figure 1) [21,22].

There are a huge number of other applications of Sharpless asymmetric dihydroxylation. For example, Sharpless AD has been found to be the main step in the total synthesis of Codonopsinine, Synargenotolides, Noneneolide and many more, which are isolated from different plant extracts and play a vital role against a number of bacterial diseases, in combating the spread of tumor cells and exhibiting high efficacy against malarial infections, respectively [23,24,25]. Our review highlights the recent applications of Sharpless asymmetric dihydroxylation for the synthesis of various natural products.

## 2. Review of the Literature

### 2.1. Synthesis of Alkaloid-Based Natural Products

#### 2.1.1. Lycorine-Type Alkaloids

Zephyranthine belongs to the class of lycorine alkaloids, which are known for their medicinal usage [26]. Members of this class have been found to interrupt the acetylcholine activity and uncontrolled division of cancer cells [27]. Zhao et al. [28] in 2021 reported the efficient synthesis of (-)-zephyranthine by utilizing two one-pot reactions. Evans’ nickel (II) catalyst **9** initiated the synthesis by reacting unsaturated β-ketoeaster **5** and nitro olefin **8** via a series of Michael addition reactions. Both reactants **5** and **8** were prepared individually. In this regard, starting material 2,2,6-trimethyl-[1,3]dioxin-4-one **3** was reacted with diethylchlorophosphite and LiHMDS followed by oxidation in the presence of hydrogen peroxide, thereby giving compound **4** in 90% yield, which was then reacted with methanol and sodium hydride in the presence of tetrahydrofuran to give compound **5** in 87% yield. Then, substituted carbaldehyde **6** was reacted with p-TSA and ethylene glycol followed by treatment with toluene and diethylether, which resulted in the synthesis of compound 7 in 90% yield. The compound **7** was then reacted with MeNO_2_ and trifluoroacetic acid to obtain compound **8** in 84% yield. For the first Michael addition, use of toluene and Triton B were considered to be effective after optimizing the various reaction conditions. This reaction resulted in 85% yield of penta-substituted cyclohexane **10**, which is highly enantioselective (90% ee) and (>20:1) diastereoselective in nature. In order to prevent the undesirable aldol reaction, the enol moiety in **10** was protected by treating with benzoylchloride (BzCl) in the presence of pyridine, which resulted in 97% yield of benzoate **11** with more than 99% enantioselectivity. The compound 11 was cyclized through one-pot reaction on treatment with HBr and acetic acid in the presence of tetrahydrofuran followed by reaction with zinc powder and hydrochloric acid, obtaining intermediate **12** in 57% yield. The resulting intermediate was treated with Comins’ reagent and LiHMDS in the presence of tetrahydrofuran, resulting in the synthesis of triflate, which was further treated with palladium acetate, formic acid and triphenylphosphine to synthesize compound 13 in 85% yield. In the last step, enantioselective synthesis of zephyranthine **14** in quantitative yield (67%) with 7.2:1 diastereoselectivity ratio was achieved by Sharpless asymmetric dihydroxylation on reaction with AD-mix-β in the presence of tetrahydrofuran or water (Figure 2).

#### 2.1.2. Muscarine Alkaloid

There are several tetrahydrofurans containing natural products including various alkaloids such as muscarine, epi-muscarine and allo-muscarine, which are extracted from *Amanita muscaria*, i.e., a mushroom species [29]. Muscarine and its derivatives are highly potent pharmacological agents that also act as acetylcholine inhibitors [30]. Owing to their wide biological importance, Gehlawat et al. [31] in 2020 reported the total synthesis of epi-mucarine alkaloid by using various reactions including Sharpless asymmetric dihydroxylation, cyclization by bimolecular nucleophilic substitution reaction, and cleavage of epoxide ring. In the first step, (p-methoxybenzyl) glycidyl ether **15** was treated with propyl lithium in the presence of boron triflouride diethyl etherate followed by reduction with lithium aluminum hydride, resulting in the synthesis of compound **16** in 86% yield. The compound **16** was further treated with tosyl chloride in the presence of dimethyl aminopyridine and triethylamine followed by reaction with osmium tetroxide and potassium hexacyanoferrate (III) via Sharpless asymmetric dihydroxylation to obtain tosyl protected diol intermediate **17**. This newly synthesized intermediate was subjected to base catalyzed reaction to prepare compound **18** in 90% yield. Compound **18** was further made to react with cerium ammonium nitrate in the presence of acetonitrile followed by treatment with imidazole, pyrrolidine and toluene. Synthesis of epi-muscarine **19** was achieved by treatment with trimethyl amine and methanol in 95% yield (Figure 3).

#### 2.1.3. Monoterpenoid Indole Alkaloid

A major class of natural products comprises monoterpenoid indole alkaloids (MIAs), which are effective against a wide range of bacterial, viral and neurological disorders [32]. These are also useful against tumor-causing agents [33,34]. Depending upon the positioning of atoms, MIAs are divided into three categories, most of which are derived from different parts of *Alstonia scholaris.* Zhang et al. in 2020 [35] proposed the total synthesis of two important MIAs, i.e., alstolarines A and B. They stated that the total synthesis could be initiated with the forerunner difforlemenine **20**, which could be subjected to reduction followed by ring opening reaction, leading to the generation of compound **21**. Then, nucleophilic substitution could result in the synthesis of compound **22**, a common precursor in the synthesis of alstolarines A and B. For the synthesis of alstolarine A, compound **22** could be subjected to dehydration followed by Sharpless asymmetric dihydroxylation to obtain compound **23**, which could synthesize target molecule **24** via cyclization of acetal. Similarly, oxidation and removal of carboxylic acid could synthesize alstolarine B **25** (Figure 4). Both compounds were tested for their acetylcholinase inhibitory potential, and molecule **25** showed average inhibitory potential with an IC_50_ value = 19.3 µM.

#### 2.1.4. Glyphaeaside Alkaloids

The glyphaeaside alkaloids are a class of iminosugars that are found in nature, as they are derived from plants, i.e., *Glyphaea brevis* [36]. These natural products belong to the family of carbohydrates whose structures are composed of many hydroxyl groups along with phenylalkyl side-chains [37]. On the basis of the arrangement of he iminosugar center, the glyphaeasides have been categorized into three classes, named A, B and C. Glyphaeaside C, which was formerly referred to as 1-deoxynojirimycin, has been found to exhibit highly efficient inhibitory activity against β-glucosidase and snail β-mannosidase. However, it was determined that the inhibitory activity of glyphaeasides is dependent on the role of side chains. Moreover, the total synthesis of glyphaeaside C revealed that its structure closely resembles that of 2,5-dideoxy-2,5-imino-D-mannitol (D-DMDP). In 2021, Byatt et al. [38] reported the multi-step facile synthesis of glyphaeaside C. The total synthesis of our target molecule initiated with the synthesis of pyrrolidine fragments from readily available starting reagent, i.e., 2,3,5-tri-O-benzyl-β-D-arabinofuranose **26**. The fragments were then reacted to give oxazolidinone a and b. In the next step, commonly available 1-allly-4-methoxy-benzene **28** was treated with boron tribromide followed by reaction with benzyl bromide, leading to the protection of hydroxyl group. Then, Grubb’s first-generation catalyst was employed to treat pyrrolidine fragment **27** with benzyl group-protected 4-allylphenol, leading to the synthesis of a racemic mixture of compound **29** in 66–76% yield. Compound **29** was then subjected to α and β type Sharpless asymmetric dihydroxylation one by one, thereby giving **30** and **31** in 87% and 90% yields, respectively. Compounds **30** and **31** were further subjected to deprotection, leading to the synthesis of four diastereomers of our target molecule. In the end, RP-HPLC was used to separate the glyphaeaside C (Figure 5).

### 2.2. Synthesis of Terpene-Based Natural Products

#### 2.2.1. Sesquiterpenoids

Englerin is isolated from *Phyllanthus engleri*, and it has been found to exhibit significant cytotoxic potential against cancer cells of kidney [39]. Moreover, englerin A is responsible for activation of protein kinase, hence regulating the glucose level [40]. Its structure is composed of seven chiral carbons along with epoxyguaine framework. Considering the significance of englerin A, various researchers have attempted to report its total synthesis. In 2020, Mou et al. [41] described an efficient synthetic route for the total synthesis of englerin A. For this purpose, **36** was oxidized in the presence of m-chloroperoxybenzoic acid, sodium bicarbonate and dichloromethane followed by reaction with PhSiH_3_ in the presence of acetone and Co catalyst to synthesize compound **37** in 87% yield. Compound **37** was then subjected to Sharpless asymmetric dihydroxylation in the presence of K_2_OsO_4,_ (DHQ)_2_PHAL and methane sulfonamide to obtain compound **38** in 59% yield. Compound 38 was further used in the synthesis of englerin A **39** (99% yield) (Figure 6).

Goyazensolide is a biologically active, naturally occurring furanoheliangolide sesquiterpenoid that plays an effective role in combating tumor cells [42]. It is isolated from plants, and its derivatives have been found to be more active against cancer cell lines, e.g., 15-deoxygoyazensolide [43]. Considering the biologically active potential of goyazensolide, Liu et al.^.^ [44] in 2021 reported the total synthesis of this natural product. In the first step of the total synthesis, compound **40** was reacted with Dess–Martin periodinane followed by treatment with trimethylsilylacetylene via Sonogashira coupling, which yielded compound **41** in 72% yield. Compound **41** was then subjected to Sharpless asymmetric dihydroxylation via (DHQD)_2_Pyr followed by protection with tert-butyldiphenylsilyl group in the presence of imidazole and dimethylaminopyridine, which gave compound **42** in 80% yield. Compound **42** then underwent a number of steps leading to the synthesis of goyazensolide **43** in 41% yield (Figure 7).

Aromatic bisabolanes belong to the class of monocyclic sesquiterpenoids, and they are obtained from different living sources such as microorganisms, plants insects, etc. [45]. Various aromatic bisabolanes have been synthesized in their respective enantiomeric forms [46]. Yajima et al. [47] in 2021 reported the total synthesis of 1,3,5-bisabolatrien-7-ol, which involved the formation of a chiral center via Sharpless asymmetric dihydroxylation. In the first step of synthesis, compound **44** was treated with vinylmagnesium bromide followed by a reaction with methanesulfonic acid in the presence of THF and water, leading to the formation of compound **45** in 82% yield. After treatment with mercaptobenzothiazole, Sharpless asymmetric dihydroxylation was carried out, which gave chiral diol compound **47** in 98% yield. Compound **47** was further subjected to oxidation with meta-chloroperoxybenzoic acid followed by Smiles rearrangement, which gave compound **48** in 58% yield with er >99:1. Hydrogenation of compound 48 in the presence of platinum oxide led to the synthesis of *S*-Enantiomer **49** of 1,3,5-bisabolatrien-7-ol in 85% yield with an enantioselective ratio of more than 99:1. Similarly, *R*-Enantiomer **51** was achieved in an enantiomeric ratio of more than 99:1 by Sharpless asymmetric dihydroxylation of compound **46** along with similar steps (Figure 8).

Through extensive research, it has been found that the compounds isolated from the plant *Phythallus engleri* possess mighty anti-cancerous potential [48]. Owing to being composed of a glycolic ester attachment, englerin A is a more promising anti-cancerous agent as compared to englerin B, which indicates that this functional group is responsible for increased cytotoxic potential [49]. There have been numerous reports covering the total synthesis of englerin-A. So, in 2021, Palli et al. [50] attempted to explain the total synthesis of englerin A by starting with commercially available limonene **52**. Limonene **52** was transformed into compound **53** by using a number of previously reported steps, followed by its treatment with methanesulfonamide, thus carrying out Sharpless asymmetric dihydroxylation to give compound **54** in 80% yield with a more than 99% diastereoselectivity ratio. Compound **54** was then subjected to protection of diol groups on treatment with 2,2-DMP and CSA followed by hydrogenation, which gave compound 55 in 82% yield. Compound 55 then underwent a number of different steps to give a mixture of oxatricyclic alcohol **56a** and **56b**. In the next step, compound **56** was then subjected to acylation followed by elimination in the presence of Burgess reagent to obtain compound **57** in 86% yield. Compound **57** was further used to obtain target molecule **58** in 86% yield (Figure 9).

#### 2.2.2. Neoclerodane Diterpene

Neoclerodane diterpene, (-)-salvinorin A is derived from *Salvia divinorum,* which is a medicinal plant [51]. This diterpene is used as a medicinal agent and employed as a treatment for stress, anxiety and pain. Owing to its wide medicinal importance, various research groups have reported different routes for its synthesis. Recently, in 2021, Zimdar et al. [52] devised an efficient strategy for the total synthesis of salvinorin. The total synthesis was initiated with the synthesis of lactone via a number of steps resulting in the generation of lactone **60** in 87% yield. Compound **60** was further reacted with osmium tetroxide and *N*-methylmorpholine *N*-oxide in the presence of acetone and water followed by its reaction with PIDA in the presence of dichloromethane, giving keto-aldehyde **61** in 88% yield. Compound **61** further underwent a number of steps to synthesize a racemic mixture of **62** and **63** in 44% and 14% yields, respectively, with more than 99% diastereoselectivity. Compound **61** was treated with chromium trichloride and iodoform in the presence of THF to obtain compound **64** in 58% yield. Compound **64** was then further subjected to a number of steps including Mitsunobu esterification and others to obtain compound **65** in 66% yield with 94% diastereoselectivity. Compound **65** was then subjected to Sharpless asymmetric dihydroxylation, leading to the synthesis of diol moiety **66** in 95% yield. Finally, the last step involved treatment with acetic acid and DBAD(di-tert-butyl azodicarboxylate), thereby giving salvinorin A **67** in 92% yield (Figure 10).

#### 2.2.3. Nor-Triterpenoids

Plants that are members of Schisandra genus are the source of propindilactone G and its derivatives, which constitute a family of polycyclic natural products [53]. Most members of this family are abundantly utilized in several medicines in China due to their high bioactive potency [54]. Propindilactone G along with its other derivatives are obtained from *Schisandra propinqua var. propinqua*, and they are structurally composed of fused cyclic rings. Owing to their frequent utilization in medicines, several research groups have devoted attention to their total synthesis. Wang et al. [55] in 2020 reported an efficient strategy for the total synthesis of propindilactone G. Total synthesis began with the Mitsunobu reaction of **68** followed by a reaction with *N*-bromosuccinimide and perchloric acid to prepare compound **69**. It was then transformed into stable chlorohydrin, which was then joined with it to prepare compound **70** via a few steps. Compound **70** was then allowed to react with mesityl chloride in the presence of triethyl amine and tert-butanol followed by its treatment with 1,2 dimethoxyethane in the presence of DBU. It was then treated with Et_3_SiH, Co(thd)_2_ in the presence of dichloroethane and oxygen, which resulted in compound **71** in high yield (75%). Starting reagent **68** was reacted via several steps, thereby giving compound 72 in 78% yield. Compound **72** was then treated with p-TSA in the presence of toluene followed by Sharpless asymmetric dihydroxylation and treatment with HF and acetonitrile to obtain compound **73** in 96% yield. Compound **73** was converted to **74** via different steps, which then underwent Sharpless asymmetric dihydroxylation to give the target molecule, i.e., propindilactone G **75** in 58% yield (Figure 11).

#### 2.2.4. Monoterpenoids

Aromatic 3,5-dimethylorsellinic acid (DMOA) is a source of many fungal monoterpenoids that constitute a variety of structures [56]. DMOA-isolated monoterpenoids, i.e., berkeyleyone and preaustinoids 1, preaustinoid 2 and preaustinoid 3, are biologically active and are highly potent against a number of inflammatory diseases [57]. (–)Berkeyleyone A structure is composed of a nonane core based on four cyclic rings. Keeping in view the medicinal importance of DMOA-derived (-)berkeyleyone A and preaustiniod 1-3, Zhang et al. [58] in 2021 described an efficient synthetic route towards their total synthesis. In this regard, 2,4,6-trihydroxybenzoic acid hydrate **76** was passed through four different steps and gave compound **77,** which was subjected to reduction followed by treatment with **78** (which was obtained by Sharpless asymmetric dihydroxylation of **85**) and later on, passing through sequential Krapcho dealkoxycarbonylation to obtain compound **79**. Compound **79** was then subjected to Wittig olefination followed by base catalyzed acylation. Then, Krapcho-type demethylation was carried out followed by protection with trimethylsilyl group, which was then allowed to undergo Mander’s reagent-catalyzed acylation. Finally, deprotection of the silyl group led to the synthesis of (-)berkeleyone A **80** in 94% yield. Berkeyleyone A underwent Dess–Martin periodinane-mediated oxidation to synthesize preaustinoid A1 **81** in 88% yield, which was then treated with boron trifluoride diethyl etherate in the presence of MeCN to synthesize preaustinoid B **83** in 86% yield, which was converted to preaustinoid B2 **84** by the addition of aqueous NaOH and absolute alcohol. Similarly, preaustinoid A **81** was subjected to oxidation in the presence of m-chloroperoxy benzoic acid to yield preaustinoid A1 **82** in 43% yield (Figure 12).

#### 2.2.5. Monoterpenoid Alcohol

The new monoterpene alcohol was first isolated from a Chinese herb, namely ‘*Mentha haplocalyx*’, by Liu and his co-researchers [59]. However, our recent studies do not correlate with their proposed structure, as it has been found to be incorrect via ^13^C and ^1^H-NMR spectroscopy. However, its structure was found to be quite similar to that of three other naturally occurring monoterpenes, namely asiasarinol, cosmosoxide B, and cis-p-menth-3-ene-1,2,8-triol [60]. This study [61] presented the total synthesis of trans-p-menth-3-ene-1,2,8-triol. The total synthesis began with the α and β-Sharpless asymmetric dihydroxylation, leading to the synthesis of compounds **87a** and **87b** in 40 and 43% yields with 33.8% and 1.2% enantiomeric excess, respectively, and similarly via route B, AD-mix-α and β leading to the synthesis of **90a** and **90b** in 76% and 91% yield with 54.5% ee and 59.4% enantiomeric excess, respectively. Compound **87a** was further treated in the presence of methyl lithium and tetrahydrofuran followed by reaction with Dess–Martin periodinane and dichloromethane, thus finally converting to the synthesis of trans **88a** (A-α) in 47% yield with 99.5% enantiomeric excess, by Luche reduction. On the other hand, other enantiomers of trans-monoterpenes were obtained by oxidation in the presence of manganese oxide followed by Luche reduction via both routes A and B. In this way, both enantiomers of the target molecule were obtained by using Sharpless asymmetric dihydroxylation (Figure 13).

### 2.3. Synthesis of Polyketide-Based Natural Products

Pladienolides A and B have been isolated from *Streptomyces platensis*, and they are known to be actively involved in the division of messenger ribonucleic acid (mRNA) [62]. Moreover, it has been discovered that pladienolide B is highly effective against the virus SARS-CoV-2 [63].Taking into account the wide pharmacological aspects of pladienolides, Rhoades et al. [64] in 2021 reported a facile and efficient route for their total synthesis. In the first step, starting reagent **91** was reacted with 3-buten-2-yl-acetate **92** in the presence of iridium catalyst and potassium phosphate to obtain compound **93** in 91% yield. Compound **93** was subjected to Sharpless asymmetric dihydroxylation, which gave compound **94** in 77% yield with 10:1 diastereoselectivity. Compound **94** was then reacted with NIS followed by its treatment with acetic anhydride, trimethylamine and DMAP to obtain compound **95** in 90% yield. Compound **96** (which was originally obtained via a number of steps) was made to react with **95** in the presence of Pd(dppf)Cl_2_, potassium phosphate and compound **97**, leading to the synthesis of target molecule **98** in 87% yield (Figure 14).

*Actinomycete streptomyces* is a source of pharmacologically important alchivemycins A and B, which are polyketides comprising highly complicated structures [65]. These naturally occurring polyketides play essential roles against a number of bacterial diseases. Their structures consist of an oxazine heterocyclic ring and 17-membered polycyclic central ring along with five optically active carbon-containing C16-C25 fragments [66]. For the first time, Liao et al. [67] in 2021 attempted to report the total synthesis of alchivemycins A and B by describing a synthetic route to obtain C16–C25 moiety. Achmatowicz rearrangement and enantioselective dihydroxylation were the main steps of their synthetic scheme. The synthetic route started with the aldol condensation by using optically active Evans’ oxazolidinone in the presence of trimethylamine and dichloromethane. Later, compound **99** was obtained by treatment with t-butyl silyl chloride in the presence of imidazole, DMAP and DCM. Compound **99** was reacted further via different steps to obtain the alkene-containing compound **100** in 65% yield. Compound **100** was then treated with p-methoxy benzoyl chloride in the presence of other reagents, which gave compound **101** in 65% yield. Compound **101** was further subjected to Sharpless asymmetric dihydroxylation in the presence of t-butanol and water to give compound **102** in 66% yield. Compound **102** was then transformed into C16-C25 fragment **103** in 80% yield, which then could be used to synthesize alchivemycins A and B (Figure 15).

Fostriecin has been isolated from *Streptomyces pulveraceus* [68]. It is a naturally occurring polyketide that plays an effective role against different cancer lines such as lung, breast and ovarian cancer [69]. There have been various reports on the total synthesis of fostriecin since its discovery. In order to contribute to the facile synthesis of this naturally occurring polyketide, Dong et al. [70] in 2020 reported a facile synthetic route for its total synthesis. In their methodology, trienyne **104** was converted to compound **105** via several steps. Compound **105** was then subjected to olefination followed by Sharpless asymmetric dihydroxylation in the presence of osmium tetroxide and (DHQ)_2_PHAL, which resulted in the synthesis of compound **106** in 82% yield. Compound **106** was then treated with pyridine, DCM and (Cl_3_CO)_2_CO followed by reduction with formic acid to obtain compounds **107a** and **107b** in 34% and 31%, respectively. Compounds **107a** and **107b** were then subjected to Sharpless asymmetric dihydroxylation one by one followed by deprotection of the hydroxyl group in the presence of potassium carbonate and ethanol to obtain compounds **108** and **109** in 82% and 27% yield, respectively. Compound **108** was then transformed to fostriecin **110** after several steps (Figure 16).

Ascospiroketal B, which is a tricyclic core-containing polyketide, is isolated from a sea-water fungus, *Ascochyta salicorniae* [71]. Hara et al. [72] in 2020 reported the total synthesis of this natural product. The total synthesis began with the preparation of α, β-unsaturated ketone **111**, which then underwent Sharpless asymmetric dihydroxylation to synthesize diol **112** in 65% yield. The diol was protected by 2,2-dimethoxypropane and toluene sulfonic acid, followed by reaction with (EtO)_2_P(O)CH_2_CO_2_Et via Horner–Wadsworth–Emmons reaction. It was followed by reduction with DIBAL-H, resulting in the synthesis of α,β-unsaturated ester **113** in 90% yield. Compound **113** was then reacted with disopropyl *L*-tartrate via Katsuki–Sharpless epoxidation to provide compound **114** in 89% yield. Over a few steps, compound **114** was converted into compound **115**. Reaction of compound **115** with TESOTf and 2,6-lutidine in the presence of DCM followed by treatment with hydrated barium hydroxide in the presence of methanol resulted in γ- lactone **116** and **117** in 65% and 22% yield, respectively. Compound **116** was then reacted via several steps that resulted in 85% yield of targeted natural product **118** (Figure 17).

Bioinspired two-phase synthesis is described for the total synthesis of many natural products. Its first phase is referred to as the ‘cyclase phase’, which involve the production of natural products requiring the least oxidation [73]. The second one is referred to as the ‘oxidation phase’, which deals with the incorporation of oxidized functional groups leading to the synthesis of natural products. The synthesis of pharmacologically and medicinally important cytochalasins is also based on this two-phase synthesis strategy, which involves the formation of central moieties in the first stage via an integrated polyketide synthase-nonribosomal peptide synthetase (PKS-NRPS) [74]. They are composed of fused cyclic ring structures containing isoindolone moiety. Owing to their biologically active nature, efforts towards their total synthesis were inevitable. Gayraud et al. [75] in 2021 reported the total synthesis of these cyclic cytochalasin natural products, i.e., aspochalasins, leading to the synthesis of trichoderone. Their synthetic approach included Sharpless asymmetric dihydroxylation, Ireland–Claisen rearrangement, Diels–Alder reaction and Suzuki–Miyaura cross-coupling reaction as the main steps. The synthesis began with the enantioselective step, i.e., Sharpless asymmetric dihydroxylation in the presence of potassium osmate and (DHQD)_2_PHAL, providing vicinal diol **120** in 81% yield with a 90:10 enantioselective ratio, which was then subjected to Swern oxidation followed by introduction of enol **121** in 86% yield. Compound **120** was then subjected to Ireland-Claisen rearrangement in accordance with Suzuki–Miyaura cross-coupling reaction to obtain compound **124** in 71% yield. Compound **124** was then allowed to react with LDA, TBSCl and TPPA, thereby giving compound **125** in 71% yield. Compound **125** was then transformed over a number of steps into naturally occurring aspachalasan intermediate **126** in 87% yield. This intermediate then moved toward the total synthesis of trichoderone A **127** (Figure 18).

Amphirionin is a straight-chain polyketide with hexahydrofuro furan rings and alkene bonds attached to it. Amphirionin-2 was discovered from marine dinoflagellates *Amphiridium sp*. [76] and has been found to exhibit highly anti-cancerous potential [77]. It is used to inhibit the uncontrolled cell division in human colon, lungs and murine cells. Due to its wide pharmacological applications, various research groups have reported the total synthesis of amphirionin-2. Saha et al. [78] in 2020 reported the stereoselective synthesis of this 10 chiral center-containing amphirionin-2 by employing Sharpless asymmetric dihydroxylation, Julia–Kocienski olefination, cycloetherification, Crimmins propionate aldol reaction and Wittig olefination. For this purpose, aspartic acid **128** was converted into **129** after a number of steps and was then treated with TESCl and trimethylamine in the presence of DMAP and dichloromethane followed by Sharpless asymmetric dihydroxylation in the presence of osmium tetraoxide and NMO, and then subjected to oxidative cleavage in the presence of NaIO_4_ and sodium bicarbonate followed by reduction, leading to the synthesis of compound **130** in 76% yield. Compound **130** was then reacted further in the presence of p-toluenesulfonyl hadrazide and triphenyl phosophine followed by its reaction with m-CPBA, leading to the synthesis of compound **131** in 85% yield. Similarly, compound **132** was treated further via Sharpless asymmetric dihydroxylation to obtain compound **133** in 78% yield. Compounds **131** and **133** were then coupled via Julia–Kocienski olefination to obtain compounds **134** and **135**. Compound **135** was further subjected to several steps, affording amphirionin-2 **136** in 82% yield (Figure 19).

Isolation of angucyclinone antibiotic (+)-PD-116740 from actinomycetes has been reported by different research groups [79]. It is effective against different cancer lines, i.e., mucus-producing glandular cells, lymph cancer cells, etc.

This antibiotic belongs to trans-9,10-dihydrophenanthrene-9,10-diol-containing natural products [80]. Owing to the biological significance of heterocycle-containing antibiotics, various research groups have reported different methodologies for their synthesis. However, Zheng et al. [81] in 2021 devised a facile route for the synthesis of PD-116740 involving Sharpless asymmetric dihydroxylation and oxidative cyclization. In the first step, quinone was subjected to sodium hydrosulfite followed by protection of the –OH group by benzyl bromide, which gave **138** in 91% yield. Component **139** was obtained by the addition of bromine followed by protection with MOMCl in 96% yield. For the synthesis of component **142**, substituted benzyl alcohol **140** was treated with TBSOTf followed by reaction with n-BuLi in the presence of DMF to give compound **141** in 70% yield. Ohira–Bestmann reagent was treated with **141**, followed by treatment with B(pin)_2_ and Cu(OTf)_2_ in the presence of acetonitrile, leading to the synthesis of compound **142** in 75% yield. Palladium catalyzed Suzuki–Miyama coupling reaction was carried out between **139** and **142** followed by asymmetric dihydroxylation in the presence of osmium tetroxide and (DHQ)_2_PHAL, which led to compound **143** in 97% yield. Protection of hydroxyl groups and Cu promoted oxidation over four steps, leading to the synthesis of target molecule **144** in 63% yield (Figure 20).

### 2.4. Synthesis of Macrolide-Based Natural Products

Marine *bugula neritina* is the main source of a class of bryostatins composed of about 21 members that are known to be highly effective against cancer-causing agents. They are also responsible for neuron transmissions and also act as stimulators of protein Kinase C (PKC). Among the 21 members of this family, total synthesis of only nine bryostatins have been reported by some research groups. Bryostatin’s structure is very complicated and, therefore, it is very difficult to synthesize. To date, only one report on the total synthesis of bryostatin has been published. Keeping in view their wide biological applications, Trost et al. [82]. in 2020 took this challenging task to synthesize structurally complex bryostatin 3 via a relatively efficient and short synthetic route as compared to the one reported earlier. They utilized Sharpless asymmetric dihydroxylation, carbonylative esterification, Yamaguchi macrolactonization, epoxidation and endo-dig cyclization in their scheme. The first step of total synthesis involved the Sharpless asymmetric dihydroxylation of **145**, yielding **146** in 84% yield. Compound **146** was then treated over a number of steps to obtain compound **147** in 76% yield. Compound **147** then underwent Sharpless asymmetric dihydroxylation in the presence of (DHQ)_2_PHAL, which gave compound **148** in 90% yield. 3-Methyl butyne **149** was treated with n-BuLi in the presence of TMEDA and diethyl ether followed by reaction with 1-bromo-2-ethoxy ethane **150** in the presence of dimethyl zinc and diethyl ether, which resulted in the synthesis of intermediate **151**. The intermediate was later transformed via numerous steps into target molecule **152** in 60% yield (Figure 21).

Among recently used microtubule stabilizing agents (MSAs), zampanolide is found to be one of the most rare and effective macrolides that possesses high potential against cancer cells owing to its covalent bonding to β-tubulin [83]. Chen et al. [84] in 2020 reported the synthesis of two new zampanolide impersonators in a number of steps involving name reactions, i.e., Sharpless asymmetric dihydroxylation, Wittig reaction, Yamaguchi esterification and Horner–Wadsworth–Emmons reaction. The first step of total synthesis involved the Wittig reaction between 2-fluorobenzaldehyde 153 and ethyl 2-(triphenyl-phosphoranylidene)acetate **154,** followed by reduction leading to the synthesis of compound 155. The chiral centers at position 17 and 18 were imported either via AD-mix-α or via AD-mix- β in the presence of respective ligand (DHQ)_2_PHAL/(DHDQ)_2_PHAL and osmium salt. Compound 156 was further made to react with MsCl followed by introduction of epoxide ring via Williamson ether synthesis. Methoxymethyl (MOM) was used as protecting group for alcohol moiety, which was further treated with iodo ether **158** to give a racemic mixture of compound **159** in 22–23% yield. Yamaguchi esterification was carried out to fuse the fragments **159** and **160**, followed by treatment with the complex of hydrogen fluoride and pyridine, which resulted in the synthesis of compound **161**. Compound **161** was made to react with Dess–Martin periodinane followed by Horner–Wadsworth–Emmons condensation in the presence of barium hydroxide catalyst. In the last step, HCl was used for the deprotection of alcohols, leading to the complete synthesis of zampanolide mimic **162** (Figure 22).

Carreira et al. [85] reported the total synthesis of formosalides A and B, which play an important role in combating cancer. These are macrolides that are isolated from the marine dinoflagellate *Prorocentrum sp.* Propargylation reaction, i.e., Krische propargylation between **163** and **164** afforded compound 165 in 56% yield with 95% ee and dr >20:1. Several steps were carried out to obtain compound **166** in 44% yield. In the next step, Sharpless asymmetric dihydroxylation and asymmetric ketene cycloaddition took place one after the other to give compound **167** in 46% yield. As a result of Weinreb ketone synthesis, Evans–Tishchenko reaction and ring-closing alkyne metathesis, compound **168** was obtained in 30% yield. In the last step, Stille coupling resulted in synthesis of two formosalides **169** (Figure 23).

Patulolide C, a macrolide consisting of a lactone ring, is isolated from *Penicillium urticae,* along with patulolide A and patulolide B isomers [86]. Various research groups have evaluated their potential against a number of bacterial, fungal and viral diseases [87]. Owing to their vital pharmacological significance, there have been continuous efforts towards the total synthesis of patulolide C. However, previous methodologies reported by different research groups were linked with several strident steps, low yields and long reaction times. Paratapareddy et al. [88] in 2020 reported the facile total synthesis of patulolide C to troubleshoot the above difficulties. For this purpose, chiral epoxide **170** was treated with 7-bromohept-1-ene in the presence of magnesium followed by protection of –OH group by benzyl bromide, affording compound **171**. Compound **171** was further subjected to Sharpless asymmetric dihydroxylation in the presence of methanesulfonamide and tert-butanol:water to obtain compound **172** in 75% yield. Compound 172 was further subjected to tosyl chloride and triethylamine followed by reduction with lithium aluminium hydride. The next step involved the treatment with tert-butyldimethylsilyl chloride to give compound **173** in 87% yield. Over two steps, diol was cleaved with NaIO_4_ followed by reaction with substituted triphenylphosphine and LiOH, which gave compound **174** in 80% yield. Further, deprotection of alcoholic group and Yamaguchi reaction led to the complete synthesis of patulolide C **175** in 68% yield (Figure 24).

### 2.5. Synthesis of Amino Acid-Based Natural Products

#### 2.5.1. Tubulysins/Amino Acid

Due to developed resistance against many anti-cancerous drugs, there has been a constant urge to discover more potent anti-cancer agents [89]. Naturally occurring Tubulysins have been found to exhibit irresistible anti-cancerous activity due to their ability to interrupt tubulin division [90]. Tubulysins consist of four amino acid chains, i.e., one naturally occurring isoleucine and three other unofficial amino acids. One of them is tubuvaline (Tuv), whose facile synthesis was recently reported by Reddy et al. [91] in 2021. The key steps of this synthesis are Sharpless asymmetric dihydroxylation and aziridine ring opening reaction. The first step of synthesis involved the treatment of **176** with sodium borohydride and iodine followed by reaction with tosyl chloride in the presence of a base. Vinyl-substituted compound **178** was obtained by treatment with vinyl magnesium bromide followed by addition of di-tert-butyl dicarbonate. In the next step, compound **178** was subjected to Sharpless asymmetric dihydroxylation, which led to the efficient synthesis of optically active vicinal diols **179** in 88% yield. The alcohol moieties were protected step by step with the help of TBDPSCl and MOMCl, respectively. Compound **180** over a number of steps afforded tubuvaline fragment of tubulysins **181** in 83% yield (Figure 25).

#### 2.5.2. Amino Acid Derivative

Most of the biologically important metabolites are obtained from microorganisms. *Microascus alveolaris strain PF1466* has been found to be a major source of new cyclodepsipeptide alveolarides A-C [92], which have been discovered to be highly effective against plant pathogens and thus are environmental friendly pest killers. They have been evaluated against a number deadly plant pests in which they exhibited efficient inhibitory activity [93]. The structure of alveolarides consists of 2,3-dihydroxy-4-methyltetradecanoic acid (DHMTDA) with a high number of carbon-containing cyclic rings. Keeping in view the eco-friendly nature of alveolarides, Saha et al. [94] in 2020 described the actual structure and total synthesis of alveolaride C by employing Sharpless asymmetric dihydroxylation, macrolactamization, Julia–Kocienski olefination, amide coupling reaction, etc. In the first step, **182** was treated with butyl lithium and NaIO_4_, which gave a diol moiety that was allowed to react with PivCl followed by protection of hydroxyl group by ter-butylsilyl group, and treatment with DIBAL-H resulted in the synthesis of alcohol **183** in 80% yield. Compound **183** was then transformed into acid derivative **184** in 72% yield by undergoing a number of steps that involved Julia–Kochienski olefination as well. Aldehyde-containing compound **185** was treated with thiazolidinethione **186** in the presence of titanium tetrachloride and DCM followed by treatment with lutidine and TBSOTf. It was further subjected to reduction that furnished **187** in 88% yield. Compound **187** was then made to react under different reagents, which led to the synthesis of another acid derivative **188** in 75% yield. Compound **189** was subjected to Swern oxidation followed by Wittig olefination, which gave **190** in 85% yield. Compound **190** then underwent Sharpless asymmetric dihydroxylation and was then reacted with 2′-DMP followed by treatment with LAH to give **191** in 90% yield. Compound **191** was then reacted further to obtain **192** in 85% yield. Compound **192** was then made to react in the presence of 6:1 of methanol:water, followed by treatment with hydrated lithium hydroxide in the presence of tetrahydrofuran:methanol:water (3:1:1) to obtain **193** in 75% yield. Similarly, **192** was reused and reacted via three different steps to obtain **194** in 84% yield. Compound **194** was then reacted over a few steps, giving alveolaride C **195** in 85 % yield (Figure 26).

### 2.6. Synthesis of Flavonoid-Based Natural Products

#### 2.6.1. DHPVs/Flavonols

*Walsura trifoliata* is the source of a flavonol containing a phenylpropanoid unit [95]. Ramana et al. [96] in 2020 reported the facile, easily accessible and efficient synthetic methodology for its total synthesis via Sharpless asymmetric dihydroxylation, Grubb-2 catalyzed RCM reaction and Wittig reaction. Total synthesis began with easily available starting material **196**, which was reacted with bromo methylbenzene in the presence of DMF and sodium hydride to obtain compound **197** in 70% yield. Compound **197** over several steps resulted in the synthesis of compound **198** in 55% yield. Compound **198** was then subjected to Sharpless asymmetric dihydroxylation to obtain diol **199** in 60% yield with 99% ee, which then afforded target molecule **200** in 85% yield over several steps (Figure 27).

There are many advantages of microorganisms that live in the digestive tract of vertebrates, and there has been extensive research on these beneficial chemical compounds extracted from the intestines of vertebrates [97]. Owing to the anti-inflammatory effects of 5-(3,4′-dihydroxyphenyl)-γ-valerolactone) DHPV, Kim et al. [98] in 2020 reported the total synthesis of both enantiomers of DHPV via cross-metathesis (CM) and Sharpless asymmetric dihydroxylation. 3,4-Dimethoxyphenylpropene **201** and methyl-4-pentenoate **202** were treated in the presence of Grubbs 2nd generation catalyst and DCM via cross-metathesis to form compound **203** in 59% yield. Compound **203** was then subjected to Sharpless asymmetric dihydroxylation and AD-mix-α and β one by one followed by immediate lactonization, which led to the synthesis of compounds **204** and **205** in 62% and 55% yields, respectively. Compounds **204** and **205** were further treated separately with hydrogen molecule in the presence of Pd(OH)_2_, followed by treatment with BBr_3_ in the presence of DCM that afforded *S*- and *R*- enantiomers of DPHV **206** and **207** in moderate yields, i.e., 60% and 62%, respectively (Figure 28).

#### 2.6.2. Isoprenylated Chalcone Skeleton/Flavonoid

Various anti-cancerous drugs are isolated from natural products such as chalcones [99]. Sanjoseolide, a natural product containing an isoprenylated chalcone skeleton, obtained from *Dalea frustecens*, has wide pharmacological significance, as it has been found to be effective against inflammation, cancer and diabetes [100]. Owing to its huge medicinal importance, Tian et al. [101] in 2020 reported an efficient method for the synthesis of sanjoseolide by carrying out Sharpless asymmetric dihydroxylation, Stille coupling and Claisen–Schmidt condensation. In the first step, 2,4-dihydroxyacetophenone **208** was treated with I_2_ and KIO_3_ followed by protection with a methoxymethyl group, leading to the synthesis of compound **209** in 90% yield. Compound **209** was then reacted with Pd(PPh_3_)_4_ and **210** followed by Sharpless asymmetric dihydroxylation, giving compound **211** in 44% yield. Compound **211** was then treated with aldehyde in the presence of potassium hydroxide, and deprotection of hydroxyl groups was carried out in the presence of HCl to obtain sanjoseolide **212** in 49% yield (Figure 29).

Various natural products are composed of a homoisoflavonoid skeleton that is found to be effective against a range of diseases. Homoisoflavonoid fragments containing brazilin and other members of its family play an important role in combating different ailments. Brazilin is highly potent against bacterial, inflammatory and cancerous diseases [102]. It also plays a huge role in protection of the liver from toxicity and reduction in the tension of blood vessels. Considering the medicinal uses of brazilin, Huang et al. [103] in 2020 reported its synthetic route involving Sharpless asymmetric dihydroxylation and Prins/Friedel Craft reaction as the main steps. The total synthesis began with reaction of 4,bromo-1,2-dimethoxybenzene **213** and magnesium in the presence of iodine, thereby giving Grignard reagent, which then reacted with methyl (2-bromomethylacrylate) **214** in the presence of lithium chloride and copper iodide, leading to the synthesis of **215** in 89% yield. Compound **215** was then further treated with DIBAL-H and diisopropyl azodicarboxylate (DIAD), which gave compound **217** in 98% yield. Then, Sharpless asymmetric dihydroxylation was carried out by using K_2_OsO_4_^.^2H_2_O, potassium hexacyanoferrate (III), (DHQD)_2_PHAL and potassium carbonate in the presence of a 1:1 t-BuOH:H_2_O mixture at 0 °C, affording chiral diol containing compound 218 in 86% yield with 98% enantiomeric excess. Compound **218** was then subjected to Parikh–Doering oxidation in the presence of sulfur trioxide followed by treatment with trifluoroacetic acid to obtain compound **219** in 85% yield, which afforded targeted compounds **220** and **221** (Figure 30).

*Cudrania tricuspidata* belongs to the family Moraceae and is widely utilized as a medicinal and therapeutic drug in Asian countries. It is used in the treatment of a number of diseases. *Cudrania tricuspidata* is a major source of flavones that are very effective against cancer cell lines, liver infection and obesity [104]. Cudraisoflavone J is also derived from *Cudrania tricuspidata* and is found to be most effective against 6-hydroxydopamine, which is responsible for the death of cells in human neuroblastoma. Keeping in view the biological and medicinal potential of cudraisoflavone J, Lu et al. [105] in 2021 attempted to describe its total synthesis along with its enantiomers via Sharpless asymmetric dihydroxylation, Claisen rearrangement and Suzuki–Miyaura coupling reaction. The first step of total synthesis involved the reaction of 2′4′6′-trihydroxyacetophenone **222** with MOMCl in the presence of DIPEA and DCM followed by treatment with phenyl bromide in the presence of potassium carbonate and acetone, which gave compound **223** in 60% yield. Compound **223** was then subjected to para-Claisen rearrangement, which afforded compound **224** in 93% yield. Compound **224** was then allowed to react with benzyl bromide in the presence of potassium carbonate followed by Sharpless asymmetric dihydroxylation in the presence of methane sulfonamide, which gave compound **225** in 67% yield. Compound **225** was then treated with methane sulfonyl chloride in the presence of pyridine followed by reaction with potassium carbonate and methanol. It was then reacted with 10% palladium in the presence of charcoal, resulting in the synthesis of compound **226,** which after several steps afforded curdaisoflavone J **227** in 15% overall yield (Figure 31).

### 2.7. Synthesis of Carbohydrate-Based Natural Products

FD-594 is a highly complicated polycyclic natural product that plays an effective role against tumor cells and a variety of bacterial diseases [106]. It is isolated from *Streptomyces species* TA-0256. The structure of FD-594 is composed of six heterocyclic fused rings containing trisaccharide units along with isochromanone and 9,10-dihydrophenanthrene-9,10-diol functionalities. Keeping in view the undeniable importance of these cyclic natural products in the pharmaceutical field, Xie et al. [107] for the first time described the total synthesis of FD-594 in 20% overall yield, which involved the usage of Sharpless asymmetric dihydroxylation and copper-catalyzed oxidative cyclization. The first step of total synthesis involved the diboration of **228** in the presence of platinum followed by its cross-coupling with 1-bromo-3,5-dimethoxy benzene **229** by using palladium diacetate and RuPhos; the and resulting product was then oxidized, which led to the synthesis of compound **230** in 70% yield with 91% enantiomeric excess. Compound **230** was then reacted further and over a few steps gave **231** in 87% yield. Xanthone fragments, i.e., **232** and **233,** were treated in the presence of n-butyl lithium and tetrahydrofuran followed by oxidation in the presence of Dess–Martin periodinane, resulting in the synthesis of compound **234,** which was then reacted under different conditions to afford compound **235** in 99% yield over five steps. Compounds **231** and **235** were then coupled via Suzuki–Miyaura coupling reaction followed by Sharpless asymmetric dihydroxylation in the presence of osmium tetroxide and dichloromethane, which gave diol **236** in 96% yield. It was then further reacted in the presence of TBSOTf and 2,6-lutidine followed by treatment with trifluoroacetic acid and dichloromethane, and the resulting product was allowed to undergo hydrogenation in the presence of palladium, which gave compound **237** in 96% yield. Compound **237** then underwent further reactions in six steps that led to the synthesis of target molecule **238** in 43% yield (Figure 32).

D-xylulose is a very crucial and sparce keto sugar that is indisputable in carrying out metabolic reactions in plant and animal cells [108]. Due to its wide importance and expensive market availability, there have been continuous efforts to devise the total synthesis of D-xylulose. Previously reported methodologies faced the problem of low yields; moreover, it is generally difficult to synthesize a moiety containing several hydroxyl groups, i.e., D-xylulose. Kalagara et al. [109] in 2020 described the total synthesis of D-xylulose by using specific synthetic approaches involving Sharpless asymmetric dihydroxylation and Wittig reaction. The total synthesis commenced with the incorporation of a protecting group followed by Swern oxidation to afford compound **240** in 76% yield. In the next step, compound **241** was treated with a protecting group followed by its reaction with bromine in the presence of carbon tetrachloride, which finally led to phosphonium bromide on treatment with triphenyl phosphine. Wittig reaction was then employed, thereby producing compound **243** in 47% yield. Compound **243** was reacted with KHCO_3_ and methanol to obtain compound **244** in 36% yield. Later on, Sharpless asymmetric dihydroxylation was carried out in the presence of osmium tetroxide, (DHQD)_2_PHAL, potassium hexacyanoferrate (III), methanesulfonamide, potassium carbonate and 1:1 t-butanol:water to obtain vicinal diols **245** in 78% yield. Finally, a deprotection strategy led to synthesis of target molecule **246** in 60% yield (Figure 33).

Iminosugars are structurally similar to carbohydrates in which oxygen is replaced by a nitrogen atom [110]. These iminosugars are abundantly found in nature and are isolated from various medicinal plants. Moreover, they are biologically active against several viral diseases, and they play an important role in glycosidase inhibition [111]. In 2020, Angelis et al. [112] devised the total synthesis of iminosugar 1,4-dideoxy-1,4-imino-D-iditol in 11% overall yield by employing Sharpless asymmetric dihydroxylation. In the first step of synthesis, compound **247** was treated with sodium hydride, tert-butyldiphenylsilyl chloride and tetrahydrofuran to obtain compound **248,** which over several steps synthesized compound **249** in 74% yield. Compound **248** was also transformed into **250**, which was reacted further through several steps to obtain compound **251** in 73% yield. Compounds **249** and **251** were then subjected to Sharpless asymmetric dihydroxylation in the presence of NMO and (DHQD)_2_AQN one by one, thereby giving **252**, **253** and **254**, **255,** respectively. Compounds **252** and **253** were then reacted further via different steps to synthesize 1,4-dideoxy-1,4-imino-D-Iditol **256** and 1,4-dideoxy-1,4-imino-D-galacitol **257** in 81% and 85% yields, respectively (Figure 34).

### 2.8. Synthesis of Lactone-Based Natural Products

#### 2.8.1. Naphthoquinonopyrano-γ-lactone

Naphthoquinonopyrano-γ-lactones are monomeric or dimeric natural products that have vast pharmacological applications. Crisamacin A, which is a dimeric naphthoquinonopyrano-γ-lactone isolated from soil bacterium *Micromonospora purpureochromogenes.* Different strategies have been proposed for the total synthesis of crisamacin due to its medicinal and biological applications [113]. Kopp et al. [114] in 2020 reported the total synthesis of crisamicin A by carrying out Sharpless asymmetric dihydroxylation, oxa-Pictet–Spengler cyclization, Heck coupling and Hartwig borylation. Total synthesis began with the generation of **262** by Heck coupling of **259** and **261,** which were prepared independently from compounds **258** and **260**. In the next step, compound **262** was subjected to Sharpless asymmetric dihydroxylation, resulting in the synthesis of compound **263** in 77% yield with more than 99% enantiomeric excess. The AD was followed by ‘simple’ oxa-Pictet–Spengler cyclization, resulting in the synthesis of compound **264** in 88% yield. Next, compound 264 was treated with compound **265** in the presence of [{Ir(µ-OMe)COD}_2_], pinacolborane and 4,4′-di-tert-butyl-2,2′-dipyridyl via Hartwig borylation to attain compound **266**, which over a number of steps afforded target molecule **267** in 67% yield (Figure 35).

#### 2.8.2. Containing α,β-Unsaturated Lactone

A diverse variety of plants and animals have been found in Australian rainforests due to the favorable and alternating environment of that area [115]. Reddell and Gardon described the isolation of a family of natural products, i.e., EBC-23 and their derivatives, from the *Cinnamonum laubati* tree of the Lauraceae family. This family of natural products, and specifically EBC-23, has been discovered to play an effective role in cytotoxic activity,. EBC-23 is highly potent against a number of human cancer cell lines with minimum side effects, as normal cells are not affected by its action [116]. Due to its irrefutable cytotoxic potential, there have been numerous reports on the total synthesis of EBC-23. In 2021, Ghosh et al. [117] attempted to describe the enantioselective total synthesis of this biologically active natural product by utilizing Sharpless asymmetric dihydroxylation and Noyori asymmetric hydrogenation in 3.8% overall yield. For this purpose, easily available 1-tetradecanal **268** was converted to β-keto ester **269** via a reported methodology and was then subjected to Noyori asymmetric hydrogenation. It was then converted to Weinreb amide, which was made to react with allyl magnesium bromide in the presence of tetrahydrofuran followed by treatment with Et_2_BOMe and sodium borohydride to obtain diol, which was then protected to yield compound **270** in 87% yield. α, β-Unsaturated ester **271** was prepared by another protocol, which was described previously. Compound **271** was further subjected to Sharpless asymmetric dihydroxylation in the presence of methane sulfonamide to obtain vicinal diol **272** in 90% yield with 95% enantiomeric excess. Compound **272** was further reacted over a number of steps to obtain oxime **273** in 81% yield. Compounds **270** and **273** were then joined in the presence of oxone, potassium chloride and water followed by their reaction with iron powder, which resulted in the synthesis of a diasteromeric mixture of β-hydroxy ketone. Its isopropylidine protecting group was detached by using Montmorillonite K_10_ in the presence of dichloroethane, leading to the synthesis of target molecule EBC-23 **274** in 42% yield (Figure 36).

#### 2.8.3. δ-Lactone Ring Containing Natural Product/ Unsaturated Fatty Acids

Most of the compounds isolated from marine organisms have been discovered to be highly effective against a range of diseases, thus playing effective roles in pharmaceutical chemistry [118]. Ieodomycins are also obtained from a marine *bacillus* species, and they have been found to be highly potent against a variety of bacterial diseases caused by Gram-positive and Gram-negative bacteria. The chemical structure of ieodomycins consists of an alkene moiety along with hydroxyl groups. Considering the significance of such biologically active natural products, Choi et al. [119] in 2020 attempted to report the total synthesis of ieodomycins. Their synthetic approach involved Sharpless asymmetric dihydroxylation, Mukaiyama lactonization, Stille coupling, Keck asymmetric allylation and Wipf’s modification as the main steps. In this regard, firstly 4-pentyn-1-ol **275** was subjected to Swern oxidation followed by Keck asymmetric allylation in the presence of (*S*)/(*R*)-BINOL, which in turn was allowed to react in the presence of ZrCl_2_Cp_2_
**278** and Me_3_Al via Wipf’s modification to obtain compounds **278** and **279** in 59% and 74% yields, respectively. Sharpless asymmetric dihydroxylation was then employed in the presence of (DHQD)_2_PYR, resulting in the synthesis of vicinal diol **280** in 52% yield. In the next step, ketal formation took place, followed by treatment with sodium cyanate, which gave compound 281 in 88% yield. Compound **281** was then further transformed to ieodomycin A **282** in 93% yield (Figure 37).

#### 2.8.4. Styryllactone

Members of the Goniothalamus genus are of huge importance because of their unparalleled usage in the medicinal field, as they are found to be highly potent against a number of diseases such as malaria, cholera, fever, etc. Styryl lactone, i.e., cardiobutanolide, has been isolated from the parts of *Goniyothalamus elegants* [120,121]. Since its isolation, researchers have contributed to the total synthesis of this natural product, which is composed of five chiral centers. Sharpless asymmetric dihydroxylation and stereoselective olefination are the main steps of the synthetic scheme. Cardiobutanolide has been known to exhibit anti-cancerous activity. In 2021, Kovacevic et al. [122] reported the total synthesis of naturally occurring and medicinally important cardiobutanolide and 3-deoxycardiobutanolide in their work. In this regard, compound **283** was treated with tetrahydrofuran and zinc dust followed by cross-olefin metathesis in the presence of 2,2-dimethoxypropane to give compound **284** in 80% yield. Compound **284** was then subjected to Sharpless asymmetric dihydroxylation in the presence of osmium tetraoxide and NMO, which resulted in the excellent synthesis of compound **285** in 100% yield. Compound **285** was allowed to react with NaIO_4_, in the presence of a mixture of methanol and water followed by Wittig olefination and then treatment with 1:1 THF:H_2_O, which gave (Z) and (E)-isomers of compounds 286a and 286b in 88% and 66% yields, respectively. Compound 287 was subjected to Sharpless asymmetric dihydroxylation AD-mix-α and AD-mix-β followed by reaction with TFA/H_2_, which demonstrated that high yield was obtained with AD-mix-α. Compound **286** was subjected to Upjohn dihydroxylation in the presence of osmium tetroxide, butanol and NMO, which led to the synthesis of cardiobutanolide compound **293** in 51% yield along with a minimum amount of 35% of by-product compound **292** (Figure 38).

### 2.9. Synthesis of Tetrahydrofyran Ring-Based Natural Products

*Artemisia caruifolia* is the major source of highly effective caruifolin A, which is used in the treatment of a number of diseases and is a major component of the Chinese drug ‘Qing Hao’. Caruifolin A’s structure is composed of a carbon-based cyclic ring system along with THF. Fernandes A. and Bethi [123] in 2020 devised the synthesis of chiral centers and tetrahydrofuran rings in the structure of caruifolin A. For the total synthesis of our target molecule, ethyl lactate **294** was treated with protecting group followed by reduction in the presence of DIBAL-H. It was further reacted with allylbromide in the presence of zinc dust, leading to the synthesis of compounds **295** and **296** in 27% and 54% yields, respectively. Later on, compound **296** underwent ozonolysis followed by Wittig olefination in the presence of **297**, thereby giving compound **298** in 79% yield. Compound **298** was then reacted with (DHQ)_2_PHAL and potassium hexacyanoferrate (III) via Sharpless asymmetric dihydroxylation, leading to the synthesis of a racemic mixture of vicinal diol **299**. Furthermore, a nucleophilic substitution reaction was carried out followed by reduction and Wittig olefination, which resulted in the synthesis of compound **300** in 65% yield. A similar pathway was adopted to obtain diastereomer of tetrahydropyran ring **304** in 61% yield (Figure 39).

Most of the medicinally important compounds have been isolated from marine dinoflagellates [124]. Similarly, formosalide A and B are obtained from marine dinoflagellates, and they have been found to be highly potent against cancer cells [125]. Their structure is composed of a 17-membered ring along with tetrahydropyran and tetrahydrofuran ring. Lu and his co-researchers reported the total synthesis of formosalide A in recent times. However, in 2020, Gajula et al. [126] reported the total synthesis of C1-C16 fragment of formosalide B. The synthesis began with the reduction of lactone **305** by using DIBAL-H in the presence of dichloromethane followed by Wittig olefination and treatment with mesylate chloride in the presence of trimethylamine, dimethylaminopyridine and dichloromethane, which gave mesylate ester **306** in 95% yield. Compound 306 was then subjected to Sharpless asymmetric dihydroxylation by using [(DHQD)_2_PHAL], potassium hexacyanoferrate (III), potassium carbonate, methane sulfonamide and osmium tetroxide to obtain tetrahyfuran-containing moiety **307** in 88% yield. Compound **307** was reacted further via several steps, leading to the synthesis of compound **308** in 90% yield. Pentane diol **309** over several steps afforded **310** in 89% yield. Compounds 308 and 310 were then reacted with lithium chloride, DIPEA and acetonitrile followed by Sharpless asymmetric dihydroxylation in the presence of [K_3_[Fe(CN)_6_], methane sulfonamide and tert-butanol to obtain diol **311** in 80% yield. Compound **311** was reacted further with PPTS and methanol to obtain target molecule **312** in 82% yield (Figure 40).

There have been a number of reported and well-known HIV-protease inhibitor drugs, e.g., darunavir, etc. However, scientists are still trying to find more biologically active HIV-inhibitor drugs due to the threatening effects of HIV. For this purpose, Ghosh et al. [127] in 2020 reported the synthesis of tetrahydrofuran ring (THF) based aminobenzothiazole containing an HIV inhibitor by employing Sharpless asymmetric dihydroxylation as a key step. This was found to be more potent than other widely known drugs. In the first step of total synthesis, compound **313** was reduced in the presence of LiAlH_4_ followed by treatment with porcine pancreatic lipase to synthesize compound **314** in 82% yield. Compound **314** was then subjected to Swern oxidation followed by treatment with HC(OMe)_3_, Bu_4_NBr_3_ and methanol, which gave compound **315** in 84% yield. Compound **315** was then subjected to Sharpless asymmetric dihydroxylation, resulting in the mixture of diols **316** and **317**, which over several steps afforded HIV-protease inhibitor drug **318** in 65% yield (Figure 41).

Annonaceous acetogenins, which are obtained from the Annonaceae plant family, have gained huge importance in medicinal chemistry owing to their therapeutic nature [128]. Muconin is the leading anti-cancerous annonaceous acetogenin, whose structure consists of tetrahydropyran and tetrahydrofuran rings. Muconin plays a vital role in combating the uncontrolled cell division in human pancreatic and breast cells [129]. Considering the wide biological applications of muconin, Sugimoto et al. [130] in 2021 described its total synthesis by applying a number of reaction steps. Acrolein **319** was treated with lauryalmagnesium bromide **320**, thereby providing **321** in 92% yield. Compound 321 was then treated with CH_3_C(OEt)_2_ and propionic acid, followed by reduction in the presence of lithium aluminium hydride and DIBAL-H, giving compound **322** in 68% yield. Compound **322** was then subjected to reaction with Ti(O-iPr)_4_ followed by Sharpless asymmetric dihydroxylation AD-mix-β in the presence of methanesulfonic acid to produce acetal **323** in 96% yield. It was then converted to epoxide **324** over six steps, in 82% yield. Compound **324** was made to react with butenyl magnesium bromide followed by cross-metathesis by using Grubb’s 2nd generation catalyst, producing compound **326** in 74% yield. Compound 326 was further allowed to react with Cl_2_Pd(MeCN)_2_, followed by its protection with MOMCl to obtain compound **327** in 96% yield. Compound 327 then underwent Sharpless asymmetric dihydroxylation, yielding vicinal diol **328**. Over six steps, compound **328** was transformed to our target molecule muconin **329**, in 82% yield (Figure 42).

Eribulin mesylate, also known as halaven, is a medicinal drug used in the treatment of breast cancer [131]. It is isolated from natural product halichondrin B, comprising 19 chiral centers incorporated in a complex molecule, i.e., polyether. It is indeed an arduous task to synthesize this compound composed of a long carbon chain (having 35 carbon atoms). Due to its wide application as an anti-cancerous agent, its total synthesis has been performed by different researchers using a number of strategies. Similarly, Senapati et al. [132] in 2021 reported the total synthesis of C14–C28, central rings of eribulin, by using Sharpless asymmetric dihydroxylation, gold-catalyzed alkynol cyclization and cross metathesis. Within this framework, tetrahydropyran and tetrahydrofuran rings are linked via a glycosidic bond. In their strategy, the first step involved the treatment of **330** with crotyl bromide in the presence of tin, leading to the synthesis of diastereoisomers. In the next step, a hydroxyl group was protected by benzyl group, thereby giving compound **333** in 76% yield. Alkyne moiety was introduced by employing the addition of hydrogen bromide, oxidation and Ohira–Bestmann reaction to give compound **334** in 95% yield. In this regard, allyl group was then introduced by reaction with allylbromide in the presence of Dess–Martin periodinane and sodium bicarbonate followed by treatment with OBr in the presence of methanol. Compound **334** was then subjected to gold catalyzed cyclization by utilizing AgSbF_6_ and Au(PPh_3_)Cl. It was succeeded by reduction in the presence of Et_3_SiH and boron triflouride diethyletherate, which resulted in the production of compound **335** in 96% yield after protection with tert-butyl silyl group. Compounds **335** and **336** were then joined through cross-metathesis followed by Sharpless asymmetric dihydroxylation in the presence of methane sulfonamide, leading to the synthesis of **337** in 71% yield. Compound **337** was then converted to our target molecule **338** in 97% yield by employing a cycloetherification strategy. Their scheme resulted in 7.2% overall yield of eribulin fragment (Figure 43).

Numerous natural products that are obtained or isolated from several marine species are generally long-chained, high molecular weight molecules [133]. Obvious examples are amphidinol 3 (AM3), maitotoxin (MTX) and brevisulcenal-F (KBT-F). Amphidinol 3 (AM3) was obtained from dinoflagellate *Amphidium krebsii*. These are known to be involved in the destruction of red blood cells and are highly effective against various fungal diseases. Their structural formulas consist of multiple hydroxyl and alkene groups along with cyclic rings (i.e., bistetrahydropyran). In 2020, Oishi et al. [134] took a challenging job to report the total synthesis of 25 chiral centers containing amphidinol 3 for the very first time by employing Sharpless asymmetric dihydroxylation, cross-metathesis, Michael-Roush asymmetric crotylation and Mitsunobu reaction. Aldehyde-containing chain 343 was synthesized via coupling asymmetric reduction and cross-metathesis of iodoolefin **339**, alkyne **340** and amide chain **341**. Another fragment, (C21-C29) **346**, was synthesized by joining acrolein **346** with terminal olefin **345** by utilizing a number of reactions including cross-metathesis and intramolecular-oxa Michael Roush asymmetric crotylation, hydroboration and Mitsunobu reaction. Furthermore, fragments **343** and **345** were combined through Julia–Kocienski olefination, Sharpless asymmetric dihydroxylation and Nishizawa–Grieco reaction to obtain the C1-C29 fragment of amphidinol **347**. Compound **347** was then joined with C32-C52 fragment **348** and C53-C67 fragment **349** through five steps, thus finally leading to the synthesis of amphidinol 3, **350** (Figure 44).

### 2.10. Miscellaneous Natural Products

Alzheimer’s disease is one of the most dangerous disease of recent times. Drugs that inhibit the coagulation of amyloid-β (Aβ) and tau protein phosphorylation are considered effective in the treatment of Alzheimer’s disease, as the accumulation of these substances is a causative factor for this fatal disease [135]. Despite the utmost need for anti-Alzheimer’s drugs, there is a shortage of long-term, effective medicines against this disease. However, it has been discovered that 4,5-dihydroxypiperines, which are obtained from *P. retrofractum*, are effective against aluminum trichloride-induced dementia. Keeping in view the pharmacological aspects of 4,5-dihydroxypiperines, Luo et al. [136] in 2021 devised the total synthesis of their stereoisomers. For this purpose, piperine 351 was subjected to AD-mix-α and AD-mix-β one by one, thus producing **352** and **353** in 44.5% yields. Compounds **352** or **353** then underwent Mitsunobu reaction separately in the presence of NbzOH, triphenylphosphine and DEAD, thereby giving compounds **354** and **355** in 41.2 and 55% yields, respectively (Figure 45).

Penostatins A and C are isolated from a marine Penicillium species, i.e., *Enteromorpha intestinalis* [137]. These members of the penostatin family are highly effective in the treatment of cancer, as their IC_50_ values against protein tyrosine phosphatase 1B have been found to be 15.87 and 0.37µM, respectively. Keeping in view the biological importance of penostatins A and C, Wang et al. [138] in 2020 reported their total synthesis. Their synthetic scheme involved the use of Sharpless asymmetric dihydroxylation, Diels–Alder reaction and Ando–Horner–Wadsworth–Emmons olefination as key steps. The synthesis began with the formation of 6-alkyl-3-hydroxy-2-pyrone **357** in 17.6% yield, which was treated further with tert-butyl-di-phenyl silyl chloride followed by its reaction with Grubbs II generation catalyst and toluene to obtain a racemic mixture of endo and exo adduct **359a** and **359b**. These two compounds were then subjected to Sharpless asymmetric dihydroxylation to obtain a mixture of diol groups **360a** and **360b** in 85% yield. Compound **360b** was protected via acetonide, followed by removal of silyl protecting group in the presence of HF and triethyl amine, and then subsequent reduction by using sodium borohydride and lithium hydride, affording compound **362** in 85% yield. Compound 362 was reacted via several steps, thus leading to the synthesis of penostatin A **363**, which could be transformed to penostatin C **364** by dehydration in 83% yield (Figure 46).

Most of the naturally occurring and medicinally important compounds contain optically active β-amino α-hydroxy acid functionality [139]. Similarly, the side chain of the widely known anti-cancer drug taxol contains chiral β-amino-α-hydroxy acid. Thus, various researchers have reported varied methodologies for its synthesis. Considering the vital importance of this moiety, Matsushima et al. [140] in 2021 reported the total synthesis of the taxol side chain. This side chain can be synthesized from demethoxy-4-epi-cytoxazone, which is a naturally occurring compound isolated from *Streptomyces* species. Cytoxazones are well known for their anti-cancer potential, as they act as cytokine inhibitors. Trichloroacetimidates were used for the incorporation of nitrogen functionality into the side chain of taxol. For this purpose, starting reagent **366** was introduced by Sharpless asymmetric dihydroxylation followed by its reaction with DBU and trichloroacetonitrile, and then cyclization was carried out to form oxazoline ring **367** in 74% yield. Oxazoline then was reacted further via several steps, leading to the synthesis of the target molecule, i.e., taxol side chain **368** in 39% yield (Figure 47).

The resistance of deadly strains of TB against various drugs is of great concern nowadays [141]. It has been found that thuggacins are highly potent against various bacterial diseases including tuberculosis. The structure of thuggacins consists of a thiazole ring and α,β-unsaturated macrolide, and they are obtained from mycobacterium *Sorangium cellulosum*. In 2021, Tomohiro et al. [142] reported the total synthesis of medicinally important thuggacin cmc-A by employing Sharpless asymmetric dihydroxylation, Suzuki–Miyaura coupling, Sonogashira coupling and Shiina macrolactonization. The initial stage of total synthesis involved the transformation of malic acid into **370** in 75% yield. Compound **370** was then subjected to reduction followed by Swern oxidation and Wittig olefination to obtain **371** in 90% yield. Compound **371** was then treated via several steps leading to the synthesis of compound **372** in 50% yield. Optically active oxazolidinone **373** was then made to react with NaHMDS in the presence of tetrahydrofuran and toluene followed by further treatment via a known methodology to obtain compound **374** in 97% yield. Compound **374** was then subjected to Swern oxidation followed by Wittig olefination, which resulted in the synthesis of **375** in 79% yield. Next, Sharpless asymmetric dihydroxylation was carried out followed by replacement of protecting group and reduction to obtain compound **376** in 96% yield. Compound **376** was then reacted further through a number of steps to give compound **377** in 100% yield. Finally, compounds **372** and **377** were reacted over different steps, leading to the total synthesis of thuggacin cmc-A 378 in 45% yield (Figure 48).

## 3. Conclusions

This review summarized the importance of Sharpless asymmetric dihydroxylation as a key step in the synthesis of different naturally occurring compounds. This methodology plays a significant role in the functionalization of olefins by reacting alkene with minute amounts of osmium tetroxide and potassium hexacyanoferrate (III), which is an optically active ligand. However, a high level of enantioselectivity is achieved by employing chiral nitrogenous ligands. The resulting chiral vicinal diols then act as synthetic intermediates in the synthesis of various medicinal and naturally occurring organic compounds. In this review, the use of Sharpless asymmetric dihydroxylation in the total synthesis of a wide range of natural products (alkaloids, lactones, terpenoids, amino acids, etc.), reported in recent years (since 2020) was described. We hope that our review will draw the attention of synthetic chemists and encourage them to make use of highly enantioselective Sharpless asymmetric dihydroxylation in their future efforts to synthesize biologically important natural products.

## Data Availability

All data is contained in the manuscript.

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
