# Peer review of "Sharpless Asymmetric Dihydroxylation: An Impressive Gadget for the Synthesis of Natural Products: A Review"

_molecules, 2023, doi:10.3390/molecules28062722_

Round 1

Reviewer 1 Report

Comments attached

Author Response

Thank you very much for peer reviewing our manuscript and we appreciate your complimentary recommendations as your comments have helped us significantly to improve the manuscript. We have carefully scrutinized the suggestions mentioned by our worthy reviewers and in accordance of reviewer’s comments, we have revised the manuscript. 

Regards

Mariusz Mojzych

Reviewer 2 Report

The manuscript is an extensive account of the usage of the ubiquitous Sharpless asymmetric dihydroxylation in the synthesis of natural products in the years 2020-current. The descriptions of the syntheses are usually short and concise. Even so, discussing most the steps rather than just the key steps, and the relevant dihydroxylation, does make for an exhausting read. The chemical schemes are consistently created with excellent attention to detail with respect to bond angles and information on the arrows. The authors also nicely highlight the key dihydroxylation step with "blue text." The language and grammar is generally high quality but does have a variety of typographical errors (some listed below).

Recommendations to improve quality:

Line 57 describes the state-of-the-art conditions for conducting the dihydroxylation. However, these conditions are not shown with a generic scheme, or structure of the ligands. Instead the reader is expected to know this knowledge and the paper proceeds to describe some classics examples that use these condtions. Maybe the conditions should incorporated into figure 1 or a new "figure 1" created.

The rotation of text in the chemdraw schemes is rarely done, but does occur. For example scheme 11 has the reagents, borontrifluoride..., described in a difficult to read manner. Also, in schemes 22, 36,...

Typos:

line 41 enantiomeric -> enantioselective

line 88; spaces between "5&" and "8via" and "&8" are missing

line 88 "The both..." should be "Both..."

line 134 "Major class..." should be "A major class..."

line 144 space missing "24vi"

line 269 space missing "70via"

line 320 and scheme 12 indicate a "Lauche reduction" which I believe should be a Luche reduction

line 356 space missing "102was"

Scheme 14 and 15 both use AD mix to conduct the dihydroxylation. However, the blue text has different descriptors. The authors should attempt to be as consistent as possible.

line 439 Extra "enter" after "This antibiotic belongs to..."

line 486 and scheme 21 involve a "mixed asymmetric dihydroxylaiton". This term and its utility could be explained better.

Scheme 24: bold issues on number 178

Scheme 25: bold issues on number 184

line 752 extraneous underline of "Cinnamonum laubati"

line 854 "ter-butanol" should be "tert-butanol"

The references are generally well-formatted. There are a few places where spaces are missing between numbers and text. Also, a few DOIs are bold.

Author Response

(The authors gave the same response as above.)
